# The Impacts of the Clinical and Genetic Factors on Chronic Damage in Caucasian Systemic Lupus Erythematosus Patients

**DOI:** 10.3390/jcm11123368

**Published:** 2022-06-12

**Authors:** Fulvia Ceccarelli, Giulio Olivieri, Carmelo Pirone, Cinzia Ciccacci, Licia Picciariello, Francesco Natalucci, Carlo Perricone, Francesca Romana Spinelli, Cristiano Alessandri, Paola Borgiani, Fabrizio Conti

**Affiliations:** 1Reumatologia, Dipartimento di Scienze Cliniche Internistiche, Anestesiologiche e Cardiovascolari, Sapienza University of Rome, Viale del Policlnico 155, 00161 Rome, Italy; fulvia.ceccarelli@uniroma1.it (F.C.); carmelo.pirone@uniroma1.it (C.P.); licia.picciariello@uniroma1.it (L.P.); francesco.natalucci@uniroma1.it (F.N.); francescaromana.spinelli@uniroma1.it (F.R.S.); cristiano.alessandri@uniroma.it (C.A.); fabrizio.conti@uniroma1.it (F.C.); 2Università UniCamillus—Saint Camillus International University of Health and Medical Sciences, 00131 Rome, Italy; cinzia.ciccacci@unicamillus.org; 3Reumatologia, Dipartimento di Medicina e Chirurgia, Università di Perugia, 06123 Perugia, Italy; carlo.perricone@unipg.it; 4Genetics Section, Department of Biomedicine and Prevention, University of Rome Tor Vergata, 00133 Rome, Italy; borgiani@med.uniroma2.it

**Keywords:** systemic lupus erythematosus, genetics, chronic damage, polymorphisms, TNFSF4, MIR1279

## Abstract

Objective: The purpose of this study was to determine the distribution of organ damage in a cohort of systemic lupus erythematosus (SLE) patients and to evaluate the roles of clinical and genetic factors in determining the development of chronic damage. Methods: Organ damage was assessed by the SLICC Damage Index (SDI). We analyzed a panel of 17 single-nucleotide polymorphism (SNPs) of genes already associated with SLE, and we performed a phenotype–genotype correlation analysis by evaluating specific domains of the SDI. Results: Among 175 Caucasian SLE patients, 105 (60%) exhibited damage (SDI ≥1), with a median value of 1.0 (IQR 3.0). The musculoskeletal (26.2%), neuropsychiatric (24.6%) and ocular domains (20.6%) were involved most frequently. The presence of damage was associated with higher age, longer disease duration, neuropsychiatric (NP) manifestations, anti-phospholipid syndrome and the positivity of anti-dsDNA. Concerning therapies, cyclophosphamide, mycophenolate mofetil and glucocorticoids were associated with the development of damage. The genotype–phenotype correlation analysis showed an association between renal damage, identified in 6.9% of patients, and rs2205960 of TNFSF4 (*p* = 0.001; OR 17.0). This SNP was significantly associated with end-stage renal disease (*p* = 0.018, OR 9.68) and estimated GFR < 50% (*p* = 0.025, OR 1.06). The rs1463335 of MIR1279 gene was associated with the development of NP damage (*p* = 0.029; OR 2.783). The multivariate logistic regression analysis confirmed the associations between TNFSF4 rs2205960 SNP and renal damage (*p* = 0.027, B = 2.47) and between NP damage and rs1463335 of MIR1279 gene (*p* = 0.014, B = 1.29). Conclusions: Our study could provide new insights into the role of genetic background in the development of renal and NP damage.

## 1. Introduction

Systemic lupus erythematosus (SLE) is a chronic autoimmune disease characterized by multifactorial pathogenesis in which genetic background and environmental factors interplay, determining disease development [1].

In recent decades, there has been a significant improvement in managing patients with SLE in terms of survival rates; however, morbidity due to organ damage remains unresolved. The assessment of accumulated SLE-related damage has been recognized as an important achievement because it is known that specific organ damage and subsequent dysfunction are significant causes of morbidity and mortality in patients with SLE [2].

The Systemic Lupus International Collaborating Clinics/American College of Rheumatology Damage Index (SDI) was developed in 1996 [3] to assess ongoing manifestations of disease activity in SLE patients and to measure irreversible damage resulting from SLE disease activity as well as its treatment and comorbidities [4,5,6]. Moreover, the presence of specific autoantibodies, such as anti-phospholipid and anti-dsDNA, could be considered predictive factors for the development of chronic damage [7,8].

The SDI is a robust instrument for quantifying damage and has been extensively validated [9]. This tool has prognostic value; in fact, many studies have shown that damage predicts morbidity and mortality [9]. For instance, a prospective study of 230 patients over 10 years of disease duration showed that early damage was associated with a higher mortality rate [10]. High SDI scores were also associated with increased economic costs and reduced health-related quality of life [11]. Risk factors for damage include older age at diagnosis, longer duration of SLE, African-Caribbean or Asian ethnicity, high disease activity at diagnosis and greater overall activity during the disease course [12]. We previously showed that machine learning models could predict the development of chronic damage and the achievement of the Lupus Comprehensive Disease Control (*LupusCDC*) [13,14]. These models have suggested that despite the control of disease activity and the absence of adverse drug events, the chronic damage progresses in some patients, meaning that there may be other risk factors such as genetic background.

During the past two decades, genome-wide association studies have been conducted to screen hundreds of thousands of single nucleotide polymorphisms (SNPs) across the genome [15]. Meta-analyses and large-scale replication/fine-mapping studies have revealed over 100 genomic loci linked to SLE susceptibility, enhancing the understanding of SLE pathogenesis at the molecular level [15,16].

Whereas most studies have looked for an association between susceptibility loci and SLE, only a few have examined the relationships between these markers and selected disease manifestations and clinical subsets or organ damage [17,18]. For example, variants of signal transducer and activator of transcription 4 (STAT4) have been associated with a more severe disease phenotype, including ischemic stroke, nephritis and increased SDI scores [19,20]. Recently, Reid and colleagues reported that high genetic risk scores were associated with a more severe SLE phenotype, renal dysfunction, and organ damage [21]. Moving from these premises, the purpose of this study was to evaluate the contribution on chronic damage development of clinical features and genetic factors in a cohort of SLE patients. Moreover, we analyzed variants of previously identified loci associated with SLE to verify their possible contribution to the development of chronic damage evaluated as specific SDI domains.

## 2. Materials and Methods

### 2.1. Study Design and Population

A cross-sectional study was executed by enrolling Caucasian adult SLE patients attending the Lupus Clinic of the Rheumatology Unit, Sapienza University of Rome (Sapienza Lupus Cohort). SLE diagnosis was performed according to the revised 1997 American College of Rheumatology criteria [22]. We limited the analysis to subjects with a minimum disease duration of five years and at least two visits per year to the Sapienza Lupus Clinic. The Ethical Committee of AOU Policlinico Umberto I, Rome, approved the study protocol. All patients signed the informed consent for the use of their clinical and laboratory data for study purposes.

### 2.2. Clinical and Laboratory Evaluation

The clinical and laboratory data for each SLE patient were collected in a standardized, computerized, electronically filled form, including demographics, past medical history with the date of diagnosis, comorbidities and previous and concomitant treatments. Antinuclear antibodies (ANA) were determined with IIF on HEp-2, anti-dsDNA with IIF on Crithidia luciliae (titer ≥ 1:10), ENA (including anti-Ro/SSA, anti-La/SSB, anti-Sm and anti-RNP) analyzed by ELISA considering titers above the cut-off of the reference laboratory, aCL (IgG/IgM isotype) analyzed by ELISA, in serum, at medium or high titers (e.g., >40 GPL or MPL or above the 99th percentile), anti-B2 glycoprotein-I (IgG/IgM isotype) analyzed by ELISA, in serum (above the 99th percentile), and lupus anticoagulant (LA), according to the guidelines of the International Society on Thrombosis and Hemostasis. Finally, C3 and C4 serum levels were determined with nephelometry. All subjects underwent blood drawing (5 mL supplemented with 0.5% EDTA) to perform genetic analysis. We registered clinical and laboratory data referring to the whole patient’s disease history.

### 2.3. Disease Activity

We assessed the disease activity according to the SLE Disease Activity Index 2000 [SLEDAI-2K] in all visits available in the three years prior to the SDI assessment [23]. We identified, in our cohort, three different patterns of disease activity according to SLEDAI-2K values, as follows: (1) patients with SLEDAI-2K ≤ 2 on all the available visits [minimal disease activity (MDA)]; (2) patients with SLEDAI-2K ≥ 4 on at least two consecutive visits [persistent active disease (PAD)]; (3) patients with at least one flare defined as an increase in SLEDAI-2K ≥ 4 from the previous visit [relapsing–remitting disease (RRD)].

### 2.4. Chronic Damage

Chronic damage was determined based on SDI at the last available examination in our center. The SDI score was calculated based on organ damage that occurred after diagnosis with SLE [3]. The SDI assesses 41 items across 12 organ systems: ocular (range 0–2), neuropsychiatric (0–6), renal (0–3), pulmonary (0–5), cardiovascular (0–6), peripheral vascular (0–5), gastrointestinal (0–6), musculoskeletal (0–7), skin (0–3), gonadal (0–1), endocrine (0–1) and malignancy (0–2). Most items were assigned 1 point if present, with 2 points possible for recurrent events and 3 points for end-stage renal disease, for a possible total score of 47. To distinguish damage from reversible disease activity, an item must be present for at least six months to be scored, irrespective of the cause. Four items of the SDI focus specifically on glucocorticoid- (GC) related adverse effects (cataracts, osteoporotic fracture, avascular necrosis, diabetes mellitus). From the sum of these items, we generated a single glucocorticoid-related SDI domain (GC-SDI) as previously described [24,25].

### 2.5. DNA Extraction and Genotyping

Genomic DNA was isolated from peripheral blood mononuclear cells using a Qiagen blood DNA mini kit. Based on literature data, we selected a panel of 17 SNPs of genes involved in immune response, autophagy and inflammation that were already described as associated with SLE [15,16,17,18]. We analyzed polymorphisms of genes linked to: innate/adaptive immune response [Toll-like receptor and type I interferon signaling: rs7574865 (STAT4), rs3027898 (IRAK1)]; T cell signaling [rs22205960 (TNFSF4)]; T and B cell signaling and interaction [rs1800872 and rs3024505 (IL10), rs4810485 (CD40); self-antigen clearance defects [rs2241880 (ATG16L1)]; autophagy [rs6568431, rs2245214 and rs573775 (ATG5)], rs13361189 and rs4958847 (IRGM)]; genes located in the HLA region [rs9469003 and rs3099844 (HCP5)] and microRNAs [rs1463335 (MIR1279), rs2431697 (MIR146a), rs531564 (MIR124A)].

Genotyping was performed with a TaqMan allelic discrimination assay (Applied Biosystems, Foster City, CA, USA) and real-time PCR. Each assay was run including samples with known genotypes previously confirmed by direct sequencing as genotype controls.

### 2.6. Statistical Analysis

The statistical evaluation was performed using dedicated software: Statistical Package for Social Sciences 13.0 (SPSS, Chicago, IL, USA) and GraphPad 5.0 (GraphPad Software, La Jolla, CA, USA). Normally distributed variables were summarized using the mean, standard deviation (SD) and nonnormally distributed variables by the median and interquartile range [IQR]. Wilcoxon’s matched pairs test and paired t-test were performed. For the univariate analysis, two groups of patients, with and without damage by SDI score, were considered. The differences between categorical variables were calculated using a chi-square test or Fisher’s exact test where appropriate. A Spearman correlation analysis was performed for measuring the correlation between variables. Two-tailed P values were reported, and P values less than or equal to 0.05 were considered significant. Odds ratios (ORs) with 95% confidence interval (CI) were calculated. A genotype–phenotype correlation analysis was performed considering the heterozygotes and variant homozygotes together (one degree of freedom). A binary logistic regression analysis (stepwise) was performed to analyze the contributions of specific SNP variants to the development of chronic damage as specific SDI domains.

## 3. Results

We analyzed 175 Caucasian SLE patients (M/F 15/160, median age at disease diagnosis 31 years, IQR 18; median disease duration 227 months, IQR 138). Table 1 summarizes the primary demographic and clinical data, laboratory features and patterns of disease activity in the whole SLE cohort, including ongoing and previous treatments. In our cohort, joint involvement was the most frequent clinical feature (89.1%), followed by skin manifestation (85.7%).

At the time of study enrollment, 105 out of 175 (60%) of SLE patients showed chronic damage in at least one organ/system (SDI ≥ 1), with a median value of 1.0 (IQR 3.0).

As expected, a significantly higher median age and median disease duration were observed in patients who had SDI ≥ 1 in comparison with patients without chronic damage [age: 54 years (IQR 14) versus 46 years (IQR 16); *p* = 0.0001; disease duration: 267 months (IQR 156) versus 183 months (IQR 108); *p* = 0.0001).

We registered a significant difference in the prevalence of some disease-associated manifestations, serological parameters and drugs prescribed between the two groups (Table 1). In detail, damage accrual was significantly associated with neuropsychiatric manifestations (*p* = 0.00001), anti-phospholipid syndrome (*p* = 0.0017) and the positivity of anti-dsDNA antibodies (*p* = 0.0099) and LA (*p* = 0.04). Concerning therapies, cyclophosphamide (CY) and mycophenolate mofetil (MMF) were more frequently prescribed in patients with chronic damage (*p* = 0.00001, *p* = 0.0058, respectively). Moreover, as is well-known, GC treatment influenced irreversible damage development. All our patients have received GC therapy, but a significantly higher proportion of SLE patients with chronic damage took glucocorticoids for a period longer than ten years (*p* = 0.00001).

Looking at disease activity, we found a similar prevalence of disease activity patterns in patients with and without chronic damage. However, when comparing the median SDI, patients with PAD showed a significant higher value in comparison with MDA patients [2.0 (IQR 4.5) vs. 1.0 (IQR 3.0); *p* = 0.04].

Figure 1 reports the SDIs for our SLE cohort. Most patients had low values [29 patients (16.6%) showed SDI = 1,26 patients (14.8%) had SDI = 2], whereas only a minority of our patients had SDI scores higher than five points. In detail, five patients (2.8%) had SDI = 6, and five patients showed SDI = 7. Only one patient, who had a disease duration of 508 months, had SDI = 10, which is the highest score registered in our cohort.

In Table 2, we show the distribution of damage according to each SDI domain. The musculoskeletal domain was the most frequently involved organ/system (46/175 patients, 26.2%), followed by the neuropsychiatric and ocular domains, detected in 43 (24.6%) and 53 (20.6%) patients, respectively.

Moving to the assessment of chronic damage related to the side effect of GC treatment, 41 patients (23.4%) developed damage in the GC-SDI domain, of whom 11 had more than one organ/system affected in this peculiar domain.

### Genotype-Phenotype Correlation Analysis

We further performed a genotype–phenotype correlation analysis to evaluate the possible associations between the above-reported polymorphisms and the development of chronic damage evaluated as specific SDI domains and items. Our study showed a potential role for variants of three different genes.

In detail, we found an association between renal damage, identified in 6.9% of patients, and TNF Superfamily Member 4 (TNFSF4) rs2205960 SNP (G > T) (*p* = 0.001). Only this genetic variant was significantly associated with renal damage, showing that individuals carrying the variant T allele (GT and TT genotypes) had a higher risk of developing this kind of damage in comparison with individuals carrying the wildtype genotype (GG) (*p* = 0.001, OR 17.0, 95% CI 2.122–136.769) (Figure 2).

Moreover, this SNP was significantly associated with the development of two specific items of renal domain: end-stage renal disease (ESDR) (*p* = 0.018, OR 9.68, 95% CI 1.136–82.527) and estimated glomerular filtration rate (GFR) <50% (*p* = 0.025) (Figure 2).

Furthermore, we found an association between the rs1463335 SNP (T > A) of microRNA 1279 (MIR1279) gene, mapping to chromosome 12q15, and the development of neuropsychiatric damage (29.1% of patients; *p* = 0.029; Figure 3A). Patients carrying the variant A allele seem to have an increased risk of developing this type of damage (*p* = 0.029, OR 2.783. 95% CI 1.081–7.165), but this polymorphism is not associated with the development of specific neuropsychiatric domain items.

The multivariate logistic regression analysis adjusted for main confounders [sex, age, disease duration, GC treatment duration, MMF and CY treatment and, aPL positivity) confirmed the association between renal damage and rs2205960 of TNFSF4 (*p* = 0.027, B = 2.47) and between neuropsychiatric damage and rs1463335 of MIR1279 (*p* = 0.014, B = 1.29).

Finally, we observed a significant association between HLA complex P5 (HCP5) rs9469003 SNP (T > C), on chromosome 6, and the GC-SDI domain (*p* = 0.028), suggesting that the variant C allele may confer an increased risk of developing damage related to the side effects of GC treatment (OR 2.6; 95% CI 1.091–6.197; Figure 3B).

## 4. Discussion

In the present study, we aimed to evaluate the contribution of genetic background to the development of chronic damage in terms of specific SDI domains.

In our cohort, 60% of SLE patients showed damage after a median disease duration of almost 19 years. In previous reports, more than 60% of patients had irreversible damage within 7 years of diagnosis of SLE [26,27]. Furthermore, our results identified demographic factors, anti-phospholipid antibodies and treatment with GC as predictors of chronic damage [5,6,11,12,24,28]. In fact, we confirm the worse prognostic effect of high age and disease duration, and we acknowledge the main role of GC treatment in determining chronic damage. Thus, most patients with SDI ≥ 1 receive GC for a cumulative period of more than 10 years and, 23.4% of them had an impairment in 1 or more items of the GC-SDI domain.

Moreover, we found a significant association between CY and MMF administration and the presence of irreversible damage. Certainly, the involvement of major organs could be a confounding factor because these drugs are generally administered to patients with more severe disease manifestations, such as proliferative nephritis or central nervous system vasculitis. Of note, treatment with CY was reported associated with higher SDIs and remained a predictor of damage [29,30].

Moving to the SDI domains in our cohort, damage more frequently involved the musculoskeletal system (26.2%), followed by neuropsychiatric and ocular involvement (24.6% and 20.6%, respectively). These results agree with several studies, conducted in patients with different ethnic backgrounds, reporting that the musculoskeletal system was the most frequently damaged in SLE patients [6,31,32,33]. Data from the Hopkins Lupus Cohort and the Toronto Lupus Cohort also showed that musculoskeletal damage accrued linearly, with osteonecrosis being the most frequent subtype, followed by deforming arthritis [31,34].

It should be considered that most of our patients had erosive arthritis, and this could be related to the fact that our research group has consistently and thoroughly focused on the presence of bone erosions, as previously reported [35].

On the other hand, renal damage was uncommon in our cohort [6.9%], in contrast to most other studies (14–32.4%) [7,28,29,31,32,33], which may be because the expression of renal disease is more aggressive in some ethnic groups (11). Although renal damage was infrequent, it was significantly associated with rs2205960 of TNFSF4 gene, which was previously associated with SLE susceptibility and lupus nephritis (LN) [17,30,36].

Investigating the genetics contribution to the development of chronic damage, we found a significant association between rs2205960 SNP of TNFSF4, the development of irreversible renal damage and two specific items of this domain [end-stage renal disease and estimated GFR < 50%]. Moreover, in our analysis, we described the correlation between neuropsychiatric damage and the rs1463335 SNP in MIR1279 gene, while rs9469003 SNP in the HCP5 gene showed an association with GC-related damage.

The TNFSF4 gene is located on human chromosome 1 and encodes the TNFSF4 protein, also known as OX40 ligand (OX40L), a cytokine of the TNF ligand family. The TNFSF4 molecule is a type II transmembrane protein, which is mainly expressed on several activated immune cells. It plays an important role in effector T-cell survival, B-cell differentiation and proliferation, cytokine production and memory cell formation [36].

In 2011, Sanchez et al., analyzing the SLE susceptibility loci in a large cohort with different ethnicities, found a significant association between renal involvement and TNFSF4 gene [17]. In agreement, Aten and colleagues detected an increased TNFSF4 expression in renal biopsies of patients with LN [37]. In 2017, two different groups verified the role of TNFSF4 in SLE and LN pathogenesis using a conditional knockout mouse system and in vivo agonist and antagonist approaches in an SLE mouse model [38,39]. Both studies suggested that the OX40/OX40L pathway contributes to lupus pathogenesis by promoting the T follicular helper cell response.

Here, we described for the first time the association between rs2205960 of TNFSF4 gene and the development of irreversible renal damage, as estimated GFR < 50% and ESDR. This result could reinforce the role of this gene in SLE-related renal involvement, suggesting different pathogenic pathways for the specific disease manifestations.

The second interesting finding of our study is the association between neuropsychiatric damage and rs1463335 of the MIR1279 gene. MicroRNAs (miRNAs) are small non-coding RNA that play important functions in cell differentiation and development, cell cycle regulation and apoptosis. Emerging evidence suggests that miRNAs have various essential roles in the normal brain and that abnormal miRNA expression contributes to neurological and psychiatric diseases such as fronto-temporal dementia, Alzheimer’s disease, Parkinson’s disease, multiple sclerosis, major depression and stroke [40,41].

Interestingly, it was reported that MIR1279 presents target sites among paralogous genes of the human tyrosine family and recognizes five target miRNAs, including PTPN12 miRNA [42]. Notably, the PTPN22 gene, which is associated with SLE and multiple sclerosis susceptibility, belongs to the same family of PTPN12 [43,44]. Thus, we can speculate that this gene could be involved in neuroinflammation.

Finally, our study also describes an association between GC-related damage and the rs9469003 of HCP5 gene. HCP5 gene (major histocompatibility complex P5), located in HLA region, is expressed primarily in immune cells; thus, it could potentially play a role in autoimmune response [45]. Polymorphisms in HCP5 gene were previously described as associated with different types of severe drug reactions, such as Steven–Johnson Syndrome and toxic epidermal necrolysis [46,47]. According to this association with drug toxicity, we evaluated the two HCP5 SNPs in relation to the development of chronic GC side effects: we found only a weak association with the rs9469003.

Certainly, our study shows some limitations. The inclusion of participants of different ethnicity is needed to further investigate the role of these genetic polymorphisms in the development of chronic damage. Another limitation of our study is the relatively small number of subjects with chronic damage in a specific SDI domain, such as renal, and we lacked data regarding cumulative prednisolone dose, which is an important risk factor for the development of organ damage. Finally, although we have evaluated several SNPs, the contribution of other genetic variants should be addressed.

However, it should be underlined that this is a monocentric cohort of patients of the same ethnicity that was strictly followed and thus was well-characterized by clinical/laboratory findings and treated according to the same therapeutic approach.

In conclusion, our study showed the association of TNFSF4 with MIR1279 polymorphisms with, respectively, irreversible renal damage and the development of neuropsychiatric damage.

Our results appear promising and possibly useful in identifying patients more prone to developing specific chronic damage. These data should be considered preliminary, and a replication study in larger cohorts is strongly recommended.

## Figures and Tables

**Figure 1 jcm-11-03368-f001:**
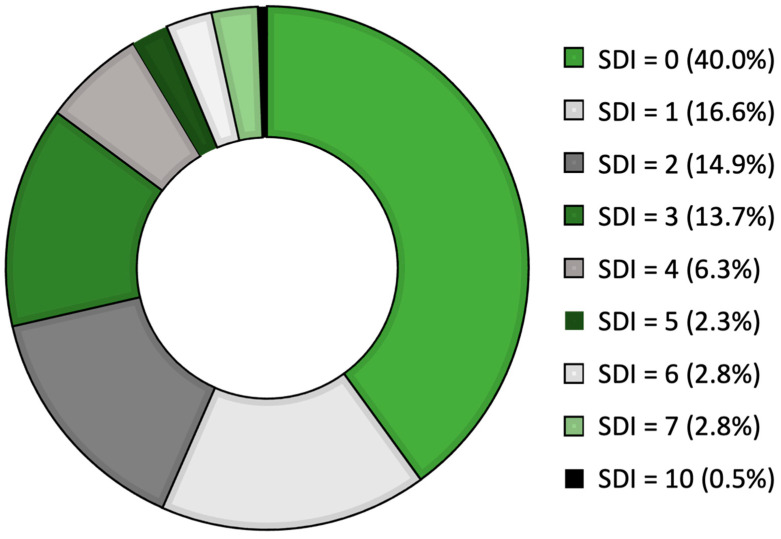
The distribution of SDIs in our SLE cohort [please read clockwise starting from the largest group with SDI = 0].

**Figure 2 jcm-11-03368-f002:**
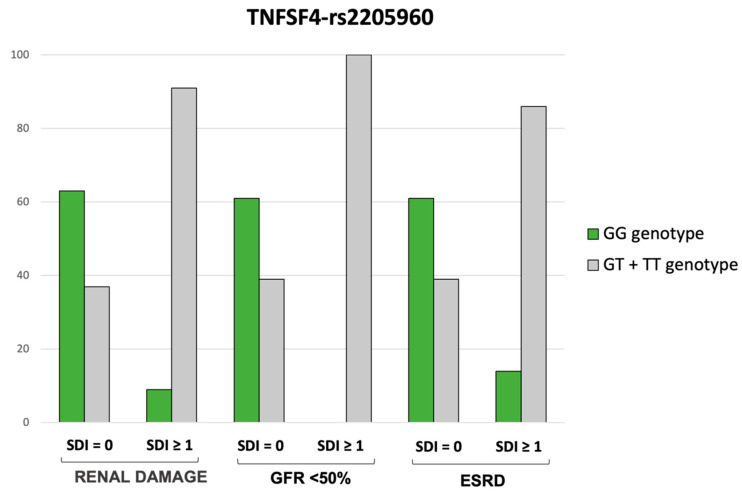
The associations between renal damage and rs2205960 of TNFSF4 [*p* = 0.001]. In addition, this SNP was significantly associated with the development of two specific items on the SDI renal domain: estimated glomerular filtration rate (GFR) <50% and end-stage renal disease (ESRD) (*p* = 0.025, *p* = 0.018 respectively).

**Figure 3 jcm-11-03368-f003:**
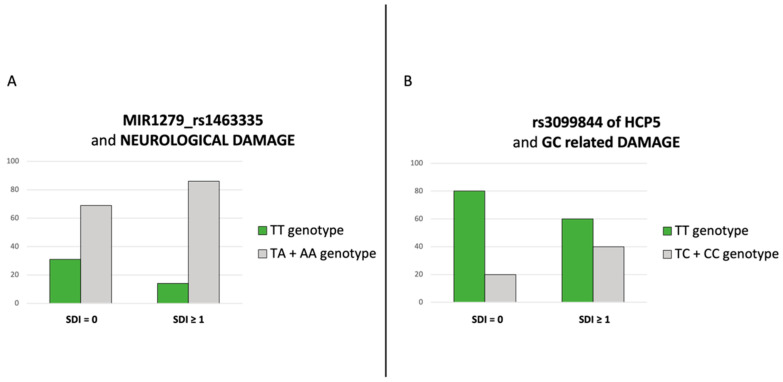
(**A**): Associations between rs1463335 of MIR1279 and the development of neuropsychiatric damage (*p* = 0.029). (**B**): rs9469003 of HCP5 locus was significantly associated with the GC-SDI domain (*p* = 0.028).

**Table 1 jcm-11-03368-t001:** Demographic and disease activity data, clinical features, serological parameters, and therapies of our SLE cohort and in the two main groups identified.

	Whole SLE Cohort N = 175	SLE Patients with SDI = 0 N = 70	SLE Patients with SDI ≥ 1 N = 105	*p* Value
M/F	15/160	3/67	12/93	n.s.
Median age–years [IQR]	31 (18)	46 (16)	54 (14)	*p* = 0.0001
Median disease duration -months [IQR]	227 (138)	183 (108)	267 (156)	*p* = 0.0001
**Disease activity patterns, n (%)**				
Minimal Disease Activity	121 (69.2)	51 (72.8)	70 (66.7)	n.s.
Persistent Active Disease	24 (13.7)	9 (12.)	15 (14.3)	n.s.
Relapsing Remitting	30 (17.1)	10 (14.3)	20 (19.0)	n.s.
**Clinical features, n (%)**				
Skin manifestation	150 (85.7)	56 (80.1)	94 (89.5)	n.s.
*Malar rash*	119 (68.0)	46 (65.7)	73 (69.5)	n.s.
*Photosensitivity*	129 (73.7)	47 (67.1)	82 (78.1)	n.s.
*Oral ulcers*	44 (25.1)	18 (25.7)	26 (26.7)	n.s.
*Alopecia*	21 (12.0)	8 (11.4)	13 (12.4)	n.s.
*Discoid rash*	16 (9.1)	7 (10.0)	9 (8.6)	n.s.
Joint involvement	156 (89.1)	59 (84.3)	97 (92.4)	n.s.
Renal involvement	67 (38.3)	22 (31.4)	45 (42.8)	n.s.
*Mesangial nephritis*	19 (10.8)	7 (10.0)	12 (11.4)	n.s.
*Proliferative nephritis*	38 (21.7)	13 (18.6)	25 (23.8)	n.s.
*Membranous nephritis*	10 (5.7)	2 (2.8)	8 (7.6)	n.s.
Hematological manifestation	101 (57.7)	39 (55.7)	62 (59.0)	n.s.
*Leukopenia*	78 (44.6)	31 (44.3)	47 (44.7)	n.s.
*Thrombocytopenia*	44 (25.1)	14 (20.0)	30 (28.6)	n.s.
*Hemolytic anemia*	10 (5.7)	5 (7.1)	5 (4.7)	n.s.
Neuropsychiatric involvement	47 (26.8)	6 (8.6)	41 (39.0)	*p* = 0.00001
*Central NPSLE*	36 (20.6)	5 (7.1)	31 (29.5)	*p* = 0.00005
*Peripheral NPSLE*	11 (6.3)	1 (1.4)	10 (9.5)	*p* = 0.009
Serositis	48 (27.4)	15 (21.4)	33 (31.4)	n.s.
*Pericarditis*	38 (21.7)	12 (17.1)	26 (24.7)	n.s.
*Pleuritis*	30 (17.1)	9 (12.8)	21 (20.0)	n.s.
Anti-phospholipid syndrome	42 (24.0)	10 (12.3)	32 (30.5)	*p* = 0.0017
**Laboratory parameters, n (%)**				
Anti-dsDNA	132 (75.4)	46 (65.7)	86 (81.9)	*p* = 0.0099
Low C3/C4 serum levels	107 (61.1)	40 (57.1)	67 (63.8)	n.s.
Anti-cardiolipin antibodies IgM/IgG	69 (39.4)	26 (37.1)	43 (40.9)	n.s.
Anti-B2-glycoprotein I antibodies IgM/IgG	37 (21.1)	11 (15.7)	26 (24.7)	n.s.
Lupus anticoagulant	43 (24.6)	12 (17.1)	31 (29.5)	*p* = 0.04
Anti-Ro/SSA	51 (29.1)	20 (28.6)	31 (29.5)	n.s.
Anti-La/SSB	21 (12.0)	7 (10.0)	14 (13.3)	n.s.
Anti-RNP	29 (16.6)	19 (18.1)	10 (14.3)	n.s.
Anti-Sm	26 (14.8)	18 (17.1)	8 (11.4)	n.s.
**Treatments, n (%)**				
Glucocorticoids [GC]	175 (100)	70 (100)	105 (100)	n.s.
GC intake ≥ 10 years	78 (44.6)	18 (24.0)	60 (57.1)	*p* = 0.00001
Hydroxychloroquine	162 (92.6)	60 (85.7)	102 (97.1)	n.s.
Azathioprine	62 (35.4)	21 (30.0)	41 (39.0)	n.s.
Cyclosporine A	39 (22.3)	11 (15.7)	28 (26.6)	n.s.
Methotrexate	58 (33.1)	20 (28.6)	38 (36.2)	n.s.
Mycophenolate Mofetil	69 (39.4)	12 (17.1)	36 (34.2)	*p* = 0.0058
Cyclophosphamide	25 (14.3)	1 (1.4)	24 (22.8)	*p* = 0.00001
Belimumab	28 (16.0)	8 (11.4)	19 (18.1)	n.s.
Rituximab	8 (4.6)	3 (4.2)	5 (4.7)	n.s.

Legend: non-significant (n.s).

**Table 2 jcm-11-03368-t002:** The distribution of damage according to the involved organ/system.

Domain N (%)	Item	Patients N (%)
Ocular 36 (20.6)	*Any cataract ever* *Retinal change OR optic atrophy*	25 (14.3) 15 (8.6)
Neuropsychiatric 43 (24.6)	*Cognitive impairment OR major psychosis* *Seizures requiring therapy for >6 months* *Cerebral vascular accident ever OR resection not for malignancy* *Cranial or peripheral neuropathy [excluding optic]* *Transverse myelitis*	19 (10.8) 7 (4.0) 10 (5.7) 18 (10.3) 1 (0.6)
Renal 12 (6.9)	*Estimated or measured GFR < 50%* *Proteinuria >3.5 g/24 h* *ESRF [regardless of dialysis or transplantation]*	4 (2.3) 1 (0.6) 7 (4.0)
Pulmonary 5 (2.8)	*Pulmonary hypertension [right ventricular prominence or loud P2]* *Pulmonary fibrosis [clinically and/or by X-ray]* *Shrinking lung [by X-ray] 0 Pleural fibrosis [by X-ray]* *Pulmonary infarction [by X-ray] OR resection not for malignancy*	2 (1.1) 4 (2.3) 0 (0) 0 (0)
Cardiovascular 15 (8.6)	*Angina OR Coronary artery bypass* *Myocardial infarction ever* *Cardiomyopathy [ventricular dysfunction]* *Valvular disease [diastolic murmur, or systolic murmur >3/6] Pericarditis OR pericardiectomy*	1 (0.6) 2 (1.1) 1 (0.6) 10 (5.7) 1 (0.6)
Peripheral vascular 5 (2.8)	*Claudication* *Minor tissue loss [pulp space]* *Significant tissue loss ever [at least loss or resection of a digit]* *Venous thrombosis with swelling, ulceration, OR venous stasis*	0 (0) 1 (0.6) 2 (1.1) 2 (1.1)
Gastrointestina l24 (13.7)	*Infarction or resection of bowel below duodenum, spleen, liver, or gall bladder ever* *Mesenteric insufficiency* *Chronic peritonitis* *Stricture OR upper gastrointestinal tract surgery ever* *Pancreatic insufficiency requiring enzyme replacement OR with pseudocyst*	24 (13.7) 0 (0) 0 (0) 0 (0) 0 (0)
Musculoskeletal 45 (26.2)	*Muscle atrophy OR weakness* *Deforming or erosive arthritis* *Osteoporosis with fracture or vertebral collapse* *Avascular necrosis* *Osteomyelitis*	8 (4.6) 27 (15.4) 13 (7.4) 5 (2.9) 0 (0)
Skin 12 (6.9)	*Scarring chronic alopecia* *Extensive scarring of panniculus other than scalp and pulp space Skin ulceration [excluding thrombosis] of more than 6 months*	2 (1.1) 2 (1.1) 8 (4.6)
Gonadal	*Premature gonadal failure*	12 (6.9)
Endocrine	*Diabetes requiring therapy regardless of treatment*	9 (5.1)
Malignancy	*Malignancy [excluded dysplasia]*	18 (10.3)

## Data Availability

The datasets generated during and analyzed during the current study are available from the corresponding author on reasonable request.

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
