# Peer review of "The Impacts of the Clinical and Genetic Factors on Chronic Damage in Caucasian Systemic Lupus Erythematosus Patients"

_jcm, 2022, doi:10.3390/jcm11123368_

Round 1

Reviewer 1 Report

This study evaluates development of chronic damage (measured by SDI), identifying demographic factors, anti-phospholipid antibodies, and treatment with GC as predictors of chronic damage. The authors also identify genetic background such as  rs2205960 SNP of TNFSF4 as a factor of development of irreversible renal damage and neuropsychiatric damage and the rs1463335 SNP in MIR1279 gene. This is a very interesting study which shed some light on genetic factors that determine chronic damage. My only concern is about the use of mycophenolate and cyclophosphamide being associated as a development of damage. I think that it is the other way around, patients with organ damage require those treatments. 

Author Response

Thanks for your suggestion. As reported in the result section cyclophosphamide and mycophenolate mofetil were more frequently prescribed in patients with chronic damage. We modified the sentence in the discussion underlying that the involvement of major organs could be a confounding factor.  In detail, we believe that we found this association because these drugs are generally administered to patients with more severe disease manifestations, such as proliferative nephritis or central nervous system vasculitis.

Reviewer 2 Report

This is an interesting study that aims to characterize systemic lupus erythematosus (SLE) patients with organ damage, as well as to provide the potential genetic background. The manuscript was very well written, structured, and comprehensible. Methods were adequately described, and results were presented in a subsequent, logical manner. Conclusions were drawn appropriately as supported by the results.

Major comments:

1.     Was there any correction for multiple testing for the genotype-phenotype correlation analysis?

2.     Regarding the lab data (ANA, anti-dsDNA, …), could the authors indicate from which point the data were obtained? The earliest collected medical record, at the time of diagnosis, the last available examination, …

3.     Is there a genotype-phenotype correlation that could explain different disease patterns (minimal disease activity, persistent active disease, relapsing-remitting)?

4.     Heterozygotes and homozygotes were classified together (e.g. GT + TT for TNFSF4-rs…) for the three genotype-phenotype correlations. Could the authors justify this approach, instead of complete breakdown of the genotypes (e.g. GG, GT, TT)?

Minor comments:

1.    Table 2: Domain N vs. per item N did not always match, could the authors re-check the data?

2.    The choice of gene panel for SNP analysis was adequately elaborated and referenced. This should, however, be acknowledged in the discussion as a limitation of the study.

3.    The sentence “The clinical and laboratory data were collected…” (lines 93-95) was duplicated.

Author Response

Referee 2

This is an interesting study that aims to characterize systemic lupus erythematosus (SLE) patients with organ damage, as well as to provide the potential genetic background. The manuscript was very well written, structured, and comprehensible. Methods were adequately described, and results were presented in a subsequent, logical manner. Conclusions were drawn appropriately as supported by the results.

Major comments:

1.Was there any correction for multiple testing for the genotype-phenotype correlation analysis?

Response

Thank you for your comment. Correction for multiple testing was not performed due to the evaluation of SNPs previously associated with SLE susceptibility, so we only test the association with specific chronic damage items. Moreover, chronic damage is not so frequent and therefore it is difficult to study specific items.  Certainly, our work could be considered as a preliminary study and larger cohorts should be analyzed.

  1. Regarding the lab data (ANA, anti-dsDNA, …), could the authors indicate from which point the data were obtained? The earliest collected medical record, at the time of diagnosis, the last available examination,

Response

Thank you for your suggestion, actually we lack to write this information in the paper. We registered clinical and laboratory data referring to the whole patient’s disease history. This sentence is now added in the material ant methods section.

  1. Is there a genotype-phenotype correlation that could explain different disease patterns (minimal disease activity, persistent active disease, relapsing-remitting)?

Response

The aim of our study was to investigated the association between genetic factors and specific organ damage. Indeed, for example, we found the association between TNFSF4 gene variant, which is involved in the pathogenesis of lupus nephritis, and the presence of irreversible renal damage. We did not perform the genotype-phenotype correlation analysis between genetic factors and different disease activity patterns because these include several clinic and laboratory features summarized by SLEDAI-2k. So, in our opinion, is it difficult to believe that a specific genetic variant could be associated to an activity’s pattern which include many features.

  1. Heterozygotes and homozygotes were classified together (e.g. GT + TT for TNFSF4-rs…) for the three genotype-phenotype correlations. Could the authors justify this approach, instead of complete breakdown of the genotypes (e.g. GG, GT, TT)?

Response

Thank you for your comment. This aspect is certainly a weakness of our study but we decided to combine together heterozygotes and homozygotes genotype do to the small number of patients with chronic damage in a specific domain. Now, we added this aspect in the discussion section.

Minor comments:

  1. Table 2: Domain N vs. per item N did not always match, could the authors re-check the data?

Response

In table 2 we reported the distribution of damage according to the SDI domain. Almost all domains include different items so some patients could develop more than one. For example, 36 patients develop damage in the ocular domain which is comped by 2 items: cataract and retinal change. Here, 4 patients develop both the items; for this reason Domain N and item N not always match.

  1. The choice of gene panel for SNP analysis was adequately elaborated and referenced. This should, however, be acknowledged in the discussion as a limitation of the study.

Response:

Thank you for your comment. Now we added this aspect in the limits of our study.

  1. The sentence “The clinical and laboratory data were collected…” (lines 93-95) was duplicated.

Response

Thank you for your comment, we have modified this mistake.
